# Antibacterial Activity of Biodegradable Films Incorporated with Biologically-Synthesized Silver Nanoparticles and the Evaluation of Their Migration to Chicken Meat

**DOI:** 10.3390/antibiotics12010178

**Published:** 2023-01-15

**Authors:** Meiriele da S. das Neves, Sara Scandorieiro, Giovana N. Pereira, Jhonatan M. Ribeiro, Amedea B. Seabra, Adriana P. Dias, Fabio Yamashita, Claudia B. dos R. Martinez, Renata K. T. Kobayashi, Gerson Nakazato

**Affiliations:** 1Laboratory of Basic and Applied Bacteriology, Department of Microbiology, Londrina State University, Londrina 86057-970, PR, Brazil; 2Center of Natural and Human Sciences, Federal University of ABC, Santo André 09210-580, SP, Brazil; 3Department of Food Science and Technology, Londrina State University, Londrina 86057-970, PR, Brazil; 4Laboratory of Animal Ecophysiology, Department of Physiological Science, Londrina State University, Londrina 86057-970, PR, Brazil

**Keywords:** nanotechnology, food packaging, foodborne pathogens, biodegradable polymer, silver migration

## Abstract

The food industry has been exploring the association of polymers with nanoparticles in packaging production, and active products are essential to increase the shelf life of food and avoid contamination. Our study developed starch-poly (adipate co-terephthalate butyl) films with silver nanoparticles produced with *Fusarium oxysporum* components (bio-AgNPs), intending to control foodborne pathogens. The bio-AgNPs showed activity against different *Salmonella* serotypes, including multidrug-resistant *Salmonella* Saint Paul and *Salmonella* Enteritidis, with minimum bactericidal concentrations ranging from 4.24 to 16.98 µg/mL. Biodegradable films with bio-AgNPs inhibited the growth of up to 10^6^
*Salmonella* isolates. Silver migration from the films to chicken was analyzed using electrothermal atomic absorption spectrophotometry, and the results showed migration values (12.94 mg/kg and 3.79 mg/kg) above the limits allowed by the European Food Safety Authority (EFSA) (0.05 mg/kg). Thus, it is necessary to improve the technique to avoid the migration of silver to chicken meat, since these concentrations can be harmful.

## 1. Introduction

Foodborne illnesses are one of the leading causes of death and hospitalization worldwide, reaching around 600 million hospitalization cases and 420,000 deaths annually [1]. According to the World Health Organization, salmonellae, one of the most critical microorganisms involved in foodborne diseases, causes an estimated 88 million infections and 123,000 deaths annually worldwide. This bacterium has over 2500 serotypes and can cause human infection through contaminated food or water [2]. Furthermore, the overuse of antibiotics in food production has contributed to the emergence of multidrug-resistant *Salmonella* strains [3].

Food contamination is directly linked to increased deaths yearly and causes a tremendous economic burden to the food industry. Thus, nanoparticles are a versatile and effective alternative for controlling microorganisms in food. Hence, the association of nanotechnology with materials science represents an alternative to the development of active packaging [4]. Additionally, active packaging with antimicrobial properties has also contributed to prolonging the shelf life of products by reducing the growth of microorganisms that cause food spoilage [5].

Nanotechnology is one of the research areas that has been growing continuously and arouses great interest, mainly because it is considered a tool in the development of many products in human health, contributing significantly to new antimicrobial research [6]. In this context, silver nanoparticles (AgNPs) have been explored for several applications due to their antimicrobial effect against bacteria, fungi, and viruses [7,8,9]. As a consequence, AgNPs implementation has become essential in developing active and bio-friendly materials, such as polysaccharide-based films for food packaging [10].

AgNPs can be obtained by chemical synthesis, although in addition to being expensive methods, they are also environmentally toxic. For this reason, the increased demand for these nanoparticles has encouraged the development of green nanotechnology, leading to AgNP synthesis with eco-friendly and low-cost methods using plant extracts and fungal proteins [10,11,12,13,14].

The development of active packaging has increased interest in an alternative to the massive use of plastic, aiming to replace conventional polymers [15,16]. However, some natural polymers do not present good physical properties, and to improve these characteristics, they are combined with other biodegradable polymers [15,16,17,18]. Furthermore, several studies have stimulated the incorporation of AgNPs in biodegradable films, demonstrating the potential to prevent food contamination [19,20].

Biodegradable films with AgNPs represent a suitable alternative for avoiding food contamination using eco-friendly plastic. However, more data on the impact of silver nanoparticles and the interaction between the nanomaterials and the consumer are needed to evaluate the security of the products. Therefore, it is essential to analyze the migration of this compound from packaging to food [20,21,22,23].

Hence, our study aimed to produce a biodegradable film containing AgNPs synthesized with Fusarium oxysporum components (bio-AgNPs) for the first time, presenting a silver migration within stipulated limits and demonstrating antibacterial activity against an important food pathogen, *Salmonella* sp.

## 2. Results

### 2.1. Characterization of Bio-AgNPs

The mean diameter of bio-AgNPs and their mean zeta potential were 81.25 nm (Figure 1A) and –36.4 mV (Figure 1B), respectively, with a polydispersity index (PDI) value of 0.296. Spherical nanoparticles were observed in a micrograph obtained by transmission electron microscopy (Figure 1C).

### 2.2. MIC and MBC Determination for Bio-AgNPs

All tested bacterial strains were inhibited by bio-AgNPs. MIC for bio-AgNPs ranged from 4.24 to 16.98 μg/mL, and MBC ranged from 4.24 to 16.98 μg/mL (Table 1).

### 2.3. Time-Kill Assay

For all three strains tested (*S.* Enteritidis ATCC 13076, *S*. Typhimurium UK-1 ATCC 68169, and *S.* Typhimurium ATCC 13311), the bio-AgNPs significantly reduced viable bacterial cells after 1 h of treatment, showing bactericidal effect (Figure 2).

### 2.4. Antimicrobial Tests Using Biodegradable Films with Bio-AgNPs

#### 2.4.1. Agar Diffusion Assay

Films with and without bio-AgNPs (Figure 3) were tested against five different bacteria. The results of the agar diffusion assay showed bacteria growth inhibition in the area that was in contact with bio-AgNPs films. A small clear zone around the square films can be observed in the images (Figure 4). This inhibition was not observed in the control samples.

#### 2.4.2. Quantification of Bacteria Inhibited by Bio-AgNPs Films

After contact with different numbers of bacterial cells, the bio-AgNPs films showed an inhibitory effect even against high cellular density. Films incorporated with bio-AgNPs inhibited the growth of *S.* Enteritidis ATCC 13076 and *S.* Typhimurium ATCC 13311 from 10^2^ to 10^6^ CFU/cm^2^ (Table 2). Growth of *E. coli* ATCC 25922, *S. aureus* ATCC 25923, and *S.* Typhimurium UK-1 ATCC 68169 were inhibited from 10^2^ to 10^5^ CFU/cm^2^. It was possible to detect bacterial growth for all bacteria tested in films without bio-AgNPs.

#### 2.4.3. Growth Curve with Bio-AgNPs Films

Films containing bio-AgNPs controlled the growth of all bacteria compared with the control (a film without bio-AgNPs), as shown in Figure 5. A comparative analysis among seven treatment times showed statistically significant differences related to CFU/mL (*p* < 0.05, multiple unpaired Student’s *t*-tests). The *E. coli* ATCC 25922 curve (Figure 5A) showed a mean difference of 2.38 ± 0.02 log (*p* < 0.05) between control and treatment at 2 h, and this difference increased to 4.52 ± 0.30 log (*p* < 0.05) in 12 h. The bio-AgNPs reduced the growth of *S. aureus* ATCC 25923 (Figure 5B) in 1.36 ± 0.04 log (*p* < 0.05) at 6 h and 2.81 ± 0.04 log (*p* < 0.05) at 12 h when compared with the control.

The activity of bio-AgNPs against *S*. Enteritidis ATCC 13076 (Figure 5C) was visualized at 0 h when the mean difference between control and treatment was 1.15 ± 0.11 log (*p* < 0.05). This antibacterial effect lasted for 12 h, in which a 0.97 ± 0.06 log (*p* < 0.05) difference was shown, but the most significant difference between control and bio-AgNPs films occurred at 6 h, a time point in which there was 3.57 ± 0.10 log (*p* < 0.05) difference.

The bio-AgNPs treatment of *S*. Typhimurium UK-1 ATCC 68169 (Figure 5D) showed a 1.53 ± 0.17 log (*p* < 0.05) reduction of growth compared to the control at 6 h, and the difference increased to 1.86 ± 0.05 log (*p* < 0.05) after 12 h. The growth of *S*. Typhimurium ATCC 13311 (Figure 5E) was affected by bio-AgNPs films. No viable cells were detected at 2 h of the test, but at 10 h, it was possible to count them. Even so, bio-AgNPs films caused a 5.79 ± 0.30 log (*p* < 0.05) reduction, and at 12 h, the mean difference between control and treatment was 4.35 ± 0.22 log (*p* < 0.05).

## 3. Analysis of Silver Migration from Bio-AgNPs Films to Chicken Meat

The results showed that silver migration from films to chicken meat occurred when the meat was stored in the refrigerator and freezer (Figure 6). Chicken placed in the refrigerator showed the highest silver concentration (12.94 ± 5.67 µg/g) after 10 days of storage. Samples in the freezer showed the highest silver concentration (5.43 ± 1.72 µg/g) after three days, and after 15 days of storage, the concentration was very similar (3.79 ± 1.45 µg/g).

## 4. Discussion

The silver nanoparticles’ biocidal effect has been explored in previous decades since the 18th century, once the potential of silver to act as an antimicrobial became known [24]. The study described in this manuscript demonstrated that bio-AgNPs act against different species of bacteria, and the versatility of AgNPs can be seen in other research that has reported their activity against fungi [25] and viruses [26].

The bactericidal activity of AgNPs found in our study has been explored in many recent studies, mainly nanoparticles obtained by green synthesis, as described in this study and previous research [27,28,29,30,31].

Our study showed that bio-AgNPs have activity against Gram-positive and Gram-negative bacteria. Loo et al. [27] reported AgNP MIC values of 3.9 μg/mL for *S.* Typhimurium and *S.* Enteritidis and MIC of 7.8 μg/mL for *E. coli,* in contrast to the results demonstrated in this study, in which the inhibition values for *S*. Typhimurium and *S*. Enteritidis were greater than the inhibition value for *E. coli.* In the research of Sinsinwar et al. [32], AgNPs exhibited MIC values of 26 µg/mL for *S. aureus*, 53 µg/mL for *E. coli*, and 106 µg/mL for *S*. Typhimurium; these were determinedly high values compared to the range we obtained using bio-AgNPs (4.24 µg/mL to 16.98 µg/mL). The AgNP MBC values found by Loo et al. [27] were 3.9 μg/mL for *S.* Enteritidis and 7.8 μg/mL for *S.* Typhimurium and *E. coli*; the same values found for MIC may in some cases determine MBC, which we also documented in our study, except for *S.* Typhimurium ATCC 13311 and the isolates *S.* Saint Paul 11, *S.* Seftenberg 16 and *S.* Seftenberg 19.

Studies that used the same bio-AgNPs tested in this work have presented MIC values that corroborate ours [25,29,33,34]. The chemical and physical characteristics of AgNPs influence their antibacterial activity. Although AgNPs with different sizes exhibit good efficacy, the antimicrobial activity increases when the size decreases, since the contact area of small particles is more extensive [9,35,36]. AgNPs also present a variation in morphology and stabilizing agent, and all these properties impact the silver nanoparticle concentration needed to eliminate the bacteria and are the reason for a considerable variation in MIC values among different studies. Furthermore, to test alternative antimicrobials, sometimes it is necessary to adapt a standard evaluation method, and researchers have applied different methods to study non-conventional antimicrobial compounds, making it difficult to compare results [31,37].

Results observed in time-kill assays demonstrate the remarkable efficacy of bio-AgNPs against *Salmonella* sp., reducing about 5 log (*p* < 0.05) of bacterial growth at 1 h treatment compared to positive controls. Numerous food pathogens have a rapid generation time, an important cause of bacteria infectivity [38], and the fast reduction in the number of bacteria can avoid infection once salmonellosis occurs after ingestion of at least 5 × 10^4^ bacteria in contaminated products [39]. The activity of a silver nanoparticle is also important to avoid cross-contamination and the spread of bacteria during food manipulation. Loo et al. [27] reported that after 1 h of treatment, AgNPs at 15.6 μg/mL killed *S.* Typhimurium, and 31.2 μg/mL was the bactericidal concentration for *S.* Enteritidis at the same time, with similar results to those reported in this study.

Analyzing the studies of AgNP activity, it is possible to observe that the concentration capable of eliminating bacteria varies even at the species level. The microorganisms present metabolic and structural differences relevant to better understanding the compound’s application. Silver nanoparticles can be used in combination with alternative or commercial antibiotics to prevent the selection of AgNP-resistant strains and to decrease the concentration needed to kill the microorganism. The efficacy of these combinations has already been reported [29,33,34] and can be explored more, especially against multidrug-resistant strains.

Our study developed an active packing using biodegradable polymer and bio-AgNPs. This proposal was made to work around a critical public health problem, the presence of pathogens in food [40,41,42]. *Salmonella* is an important foodborne pathogen, with many infections yearly, resulting in expressive medical care costs [43]. The contamination caused for this bacterium can occur in different steps of food production [2] and active packing is an excellent strategy to avoid consumer contamination. In our study, the bio-AgNPs presented antimicrobial activity against multidrug-resistant *Salmonella* sp. The MIC and MBC values were similar to the sensitive strains.

Several silver-based products have been developed, especially materials containing silver nanoparticles [8,44,45]. Active packaging is a concept explored to increase the shelf-life of food and decrease the risk of contamination [46]. Environmental problems created by the massive use of plastics in the industry demonstrate the necessity to expand biodegradable polymer studies.

According to different methodologies, such as disk diffusion or broth quantitative assays, the biodegradable film containing bio-AgNPs developed in our study showed antibacterial activity for all bacteria tested. The agar diffusion assay showed that bacteria inhibition occurred in the area of contact with AgNP films, and a small clear zone was also observed around the film. Abreu et al. [47] reported the inhibition of bacteria in contact areas with AgNPs films, although no inhibition halo was observed. Small or absent inhibition halo was also noted in tests with silver nanoparticles in liquid suspension plated directly on the bacterial lawn in agar plates [27,48], showing that the film polymer does not influence the presence or size of the inhibition halo. Silver nanoparticles show difficulty in migrating in agar plates, and this is reflected in the varied results reported in the literature [37]. Quantitative tests, as used in this work, are a great alternative to analyzing films, considering that they give more information to discuss and understand how this product can act.

The release of AgNP from the polymer matrix is crucial to act against microorganisms, mainly in a liquid environment, once contact with the cell membrane is essential to killing bacteria. Enumeration of bacteria inhibited by bio-AgNPs films was performed in nutrient broth and showed 5 logs of inhibition growth for *E. coli* ATCC 25922, *S. aureus* ATCC 25923, and *Salmonella* Typhimurium UK-1 ATCC 68169, and 6 logs for *Salmonella* Enteritidis ATCC 13076 *Salmonella* Typhimurium ATCC 13311. Results suggest that bio-AgNP film can avoid an infection caused by *Salmonella* sp. because it eliminates more bacteria (10^5^ and 10^6^) than is needed to cause an infection (>5 × 10^4^).

The bacterial growth curve results with bio-AgNP films showed significant (*p* < 0.05) log reduction compared to the control, even in high bacterial concentrations. More studies about bacterial kinetics associated with the film’s antibacterial activity are needed, as well as the establishment of standard techniques for product tests.

The silver nanoparticle mechanism of action is still studied because the exact antibacterial activity is yet unknown. Researchers have shown that silver nanoparticles present many targets in bacteria. Bondarenko et al. [49] showed the importance of cell-NP contact to AgNP activity. However, DNA interaction [50], free radical generation [51], ROS formation [52], and interaction with sulfur-containing proteins and enzymes [49,53,54] also have a significant role in the bio-AgNP mechanism of action. Scandorieiro et al. [55] demonstrated that the same bio-AgNP used in this work increased the membrane permeability of bacteria (cytoplasmic content leakage), and caused oxidative stress (ROS, lipid peroxidation) and ultrastructural damage (membrane and cell wall damage). These multiple targets explain that the highest activity of silver nanoparticles is observed in the log phase of growth, a phase of high metabolic activity.

The Ag^+^ release from silver nanoparticles is an important step in this compound’s antimicrobial activity [56]. However, it is also a concern when the AgNPs are applied in materials that have contact with the human body or food. The migration of silver from silver-based materials has been studied. Our research showed the presence of an average of 12.94 mg of silver per kg of chicken meat stored for 10 days in a refrigerator and 3.79 mg/kg stored for 15 days in a freezer. Our results showed a silver tendency to migrate under high temperatures, which corroborates with other studies [21,57].

Abreu et al. [47] reported an overall migration of 42.9 mg of substances (Al, Ca, Fe, and Ag) per kg of food simulant from silver nanoparticles/starch nanocomposite to food simulant. Echegoyen and Nerin [57] reported silver migration values ranging from 1 to 31 mg/cm^2^ from commercial plastic bags to food simulants, showing a tendency for silver migration. Gallocchio et al. [58] reported no migration of silver from the commercial plastic bag to chicken meatballs. However, it did not affect the bacteria tested, suggesting that migration is vital to antimicrobial activity. The migration values obtained in our study (12.94 mg/kg and 3.79 mg/kg) were within the limit determined by European directives on food packaging (EN1186-1, 2002, Regulation N° 10/2011) [59], which is 60 mg (substances)/kg (foodstuff or food simulant) of migration from plastic materials.

However, according to the European Food Safety Authority (EFSA), the specific silver limit is 0.05 mg/kg of food [60], and therefore the silver migration exceeded the limit. For this reason, more data about the effects of bio-AgNPs on humans are needed, as well as the synergistic combination of bio-AgNPs with other substances to reduce the concentration necessary to inhibit bacterial growth [61].

Evaluating silver nanoparticle migration is necessary since the cytotoxicity of AgNPs is still unclear. Scandorieiro et al. [55] used bio-AgNPs synthesized by the same protocol tested in this work in a cytotoxicity test with red blood cells (RBC). Their results demonstrated that the CC50 was 121.9 µg/mL (715.92 μM), above the values with antibacterial activity found in this study, which were in the range of 4.24 μg/mL (24.88 μM) to 16.98 μg/mL (99.72 μM). Wang et al. [62] also reported that cell damage was dose-dependent; therefore, in any product development it is essential to analyze the size and concentration of AgNPs used.

Another study that used nanoparticles synthesized from *F. oxysporum* [63] demonstrated that concentrations of 1.5–50 µM (0.26–8.51 μg/mL) did not affect the viability of RAW 264.7 cells, while the concentration of 100 µM (17.02 μg/mL) reduced cell viability to 75.05%. Machado et al. [64] also demonstrated that toxicity is dose-dependent in a model with HeLa cells, with a 17% and 31% reduction in cell viability with 9 and 12 uM (1.53 and 2.04 μg/mL) of AgNPs, respectively. Filon et al. [65] observed that the skin penetration of nanoparticles is size-dependent; nanoparticles ≤4 nm can penetrate the intact skin, while nanoparticles larger than 45 nm cannot penetrate nor permeate through the skin.

Lima et al. [66] documented the difficulty of correlating the assessment of genotoxicity in cell culture when experiments are performed on different strains and treated with different nanoparticles. However, smaller particles are more toxic, and biologically synthesized silver nanoparticles tend to be larger [65], and therefore their penetration capacity through the skin is reduced. In our study, bio-AgNPs have an average diameter of 81.25 nm, larger than the size described as able to penetrate or permeate the intact skin. Besides this, bio-AgNPs appear to be less genotoxic than other chemically synthesized silver nanoparticles [66], which can be considered an advantage.

## 5. Materials and Methods

### 5.1. Bacterial Strains

The antimicrobial assays were performed using reference strains from the American Type Culture Collection (ATCC) provided by the Laboratory of Basic and Applied Bacteriology of Londrina State University (Londrina, PR, Brazil) and 19 *Salmonella* isolates from poultry farms provided by the Laboratory of Genomics and Bacterial Molecular Biology of the University of Campinas (Campinas, SP, Brazil). The antimicrobial susceptibility profile of isolates is shown in Appendix A. The standard bacterial strains were as follows: *Staphylococcus aureus* ATCC 25923, *Escherichia coli* ATCC 25922, *Salmonella enterica* serovar Enteritidis ATCC 13076, and *Salmonella enterica* serovar Typhimurium ATCC 1331. *Salmonella enterica* serovar Typhimurium UK-1 ATCC 68169 was provided by Dr. Roy Curtiss III, Arizona State University, Tempe, AZ. The bacterial strains were stored at −80 °C in Brain-Heart Infusion Broth (BHI, Acumedia, Lansing, MI, USA) with 25% glycerol (Merck, Rahway, NJ, USA).

### 5.2. Biogenic Silver Nanoparticles

Biogenic silver nanoparticle synthesis was according to a previously established method [67]. The principle of this method is the production of bio-AgNPs after silver nitrate reduction by *Fusarium oxysporum* enzymes (strain 551), obtained through the collection of cultures from the Laboratory of Molecular Genetics of the University of São Paulo (ESALQ-USP–Piracicaba, SP, Brazil). *F. oxysporum* was cultivated on malt extract agar 2% (Difco^®^, Tucker, GA, USA) containing 0.5% (*w*/*v*) yeast extract (BD, Franklin Lakes, NJ, USA) for 7 days at 30 °C. After growth, fungal biomass was scraped off the plate using a sterile loop, added to distilled water at a concentration of 0.1 g/mL, and incubated with agitation (150 rpm) at 30 °C for 72 h. Afterward, the suspension components were separated through vacuum filtration with a qualitative filter having an average pore size of 4 to 12 mm. Silver nitrate (Sigma Aldrich^®^, St. Louis, MO, USA) at 10 mM was added to the filtrate, and the system was incubated for 15 days at 30 °C in the dark. After bio-AgNP preparation, diameter and zeta potential were determined using ZetaSizer NanoZS (Malvern, UK) at the University of Campinas, and their shape was observed using a transmission electron microscope (model JEM-1400, Jeol, Tokyo, Japan).

### 5.3. Determination of Inhibitory and Bactericidal Concentrations of Bio-AgNPs

Minimum inhibitory concentration (MIC) was determined in triplicate by broth dilution method in 96-well plates (KASVI, Campinas, SP, Brazil), according to the Clinical and Laboratory Standards Institute guidelines [68], for five reference strains being three of *Salmonella*, one of *E. coli* and one of *S. aureus*, and 19 *Salmonella* isolates. The tested concentrations of bio-AgNPs ranged from 1.06 to 67.92 μg/mL. Mueller-Hinton broth (MHB, Difco^®^, Tucker, GA, USA) alone and with bio-AgNPs was used as the negative control and sterility control, respectively, and the strains used for the MIC determination were grown in MHB with and without bio-AgNPs for tests and positive control, respectively. The bacteria tested were cultivated in Mueller-Hinton agar (MHA, Difco^®^, Tucker, GA, USA) at 37 °C for 24 h. Each bacterial strain was suspended in sterile saline and adjusted to 1.5 × 10^8^ colony-forming units (CFU)/mL using the McFarland scale. After the bio-AgNPs plus MHB were added to the microplate, the bacterial suspensions were diluted in MHB (1:100) and inoculated into the plate at a final density of 5 × 10^5^ CFU/mL. The plates were incubated at 37 °C without agitation, and after 24 h, the lowest concentration of bio-AgNPs that inhibited visible growth was defined as MIC. The minimal bactericidal concentration (MBC) was determined for all bacteria tested by plating 10 μL from the broth dilution MIC and above concentrations on MHA, and the lowest concentration that killed ≥99.9% of bacteria after the treatment (or the concentration showing no growth on MHA) was defined as MBC [69].

### 5.4. Time-Kill Assay of Bio-AgNPs

The kinetics of bio-AgNPs-treated bacterial populations were studied by the viable cell count method with the final construction of a time-response growth curve [70]. The bacteria sample was prepared as previously described in the antibacterial activity assay. Each culture was inoculated at a final cell density of 5 × 10^5^ CFU/mL in two microcentrifuge tubes, one that included a mi×ture of bio-AgNPs and MHB and the other with MHB used as growth control. Another tube with only MHB was used for sterility control. These tubes were incubated at 37 °C, and aliquots of 10 μL were collected at seven different times of incubation (0 h, 1 h, 2 h, 4 h, 7 h, 10 h, and 24 h), serially diluted in saline, and plated on MHA. The CFU/mL was determined after growth at 37 °C for 18 h. Three bacterial strains were used in this assay (*S.* Enteritidis ATCC 13076, *S.* Typhimurium ATCC 13311, and *S.* Typhimurium UK-1 ATCC 68169), and the strains were treated with bio-AgNPs at 16.98 µg, 8.49 µg, and 16.98 µg, respectively.

### 5.5. Preparation of Biodegradable Films with Bio-AgNPs

The suspension of nanoparticles was transformed into powder by freezing and lyophilization to incorporate bio-AgNPs in biodegradable films. For the lyophilization process, a lyophilizer (Model LP3, JOUAN, Winchester, VA, USA) was used for 48 h at −30 °C. Biodegradable films were produced using 30% (*w*/*w*) of poly (butylene adipate co-terephthalate) (PBAT) (Ecoflex^®^, BASF, Ludwigshafen, Germany) and 51.10% (*w*/*w*) of starch. For the silver-containing films, 17.40% (*w*/*w*) of glycerol and 1.50% (*w*/*w*) of lyophilized bio-AgNPs were used. For the control films, 18.90% (*w*/*w*) of glycerol alone was used. The ingredients of the formulations were mixed manually and extruded in a pilot single-screw extruder (BGM, model EL-25, São Paulo, SP, Brazil) to produce pellets. The extruder had a screw diameter (D) of 25 mm, a screw length (L) of 750 mm (L/D ratio of 30), five heating zones, and a die with six 2 mm holes, and the Keg temperature was set to 90/120/120/120 °C with a screw speed of 35 rpm. The pellets were extruded in the same extruder to produce three-layer films by blow extrusion, with the same operating conditions but a different die specifically for balloon formation.

### 5.6. Antimicrobial Test Using Biodegradable Films with Bio-AgNPs

#### 5.6.1. Agar Diffusion Assay

This assay was performed based on the disk diffusion method recommended by the Clinical and Laboratory Standards Institute (Clinical and Laboratory Standards Institute, 2013) [69], with modifications. Bacterial strains were grown in MHA at 37 °C for 24 h. After this incubation time, each bacterial strain was suspended in sterile saline, and the concentration was adjusted to 1.5 × 10^8^ CFU/mL using the McFarland scale. The bacterial suspension was spread in MHA plates, and a square (1 cm^2^) of the films with and without bio-AgNPs was put in contact with *E. coli* ATCC 25922, *S. aureus* ATCC 25923, *Salmonella* Enteritidis ATCC 13076, *Salmonella* Typhimurium UK-1 ATCC 68169, and *Salmonella* Typhimurium ATCC 13311. Plates were incubated at 37 °C for 24 h, and growth was observed.

#### 5.6.2. Qualitative Measurement of Bacteria Inhibition by Bio-AgNPs Films

Bacterial strains, *S. aureus* ATCC 25923, *E. coli* ATCC 25922, *S.* Enteritidis ATCC 13076, *S.* Typhimurium ATCC 13311, and *S.* Typhimurium UK-1 ATCC 68169, were inoculated in nutrient broth (Acumedia, Lansing, MI, USA, USA), and incubated at 37 °C for 24 h. Cultures reached a concentration of 10^9^ CFU/mL. After the growth, the bacterial strains were serially diluted in a saline 10-fold solution up to 10^−7^, and dilution was plated in nutrient agar to confirm the total CFUs. A volume of 10 µL of each dilution was placed on the film’s square with and without bio-AgNP and stored in a sterile Petri dish at room temperature for 24 h. The initial cellular density in contact with films ranged from 10^6^ CFU/cm^2^ to 1 CFU/cm^2^. After contact with bacteria, film samples were inoculated in nutrient broth and placed at 37 °C for 24 h. Bacterial growth was confirmed by samples plating in agar medium, *E. coli* in MacConkey Agar (Acumedia, Lansing, MI, USA, USA), *S. aureus* in Mannitol Salt Agar (Difco^®^, Tucker, GA, USA), and *Salmonella* sp. in SS Agar (Difco^®^, Tucker, GA, USA).

#### 5.6.3. Bacterial Growth Curve in Contact with Bio-AgNPs Films

This assay was performed to observe bacterial population kinetic when they were in contact with the bio-AgNPs film. Five different bacteria were used, including *E. coli* ATCC 25922, *S. aureus* ATCC 25923, *S.* Enteritidis ATCC 13076, *S.* Typhimurium UK-1 ATCC 68169, and *S.* Typhimurium ATCC 13311. 10 µL of the bacterial strain (10^9^ CFU/mL) was placed on the film’s square with and without bio-AgNPs and stored in a sterile Petri dish at room temperature for 24 h. After 24 h of contact between the film and bacteria, the film squares were inoculated in nutrient broth (1% (*m/v*) peptone, 1% (*m/v*) meat extract, 0.1% (*m/v*) glucose, and 0.5% (*m/v*) sodium chloride), and incubated at 37 °C for 12 h, with constant stirring at 160 rpm. Aliquots of 10 μL were collected every 2 h, serially diluted, and plated on the nutrient agar with post-incubation at 37 °C for 24 h. The CFU was determined, and a time x viable cells curve was constructed.

### 5.7. Analysis of Silver Migration from Bio-AgNPs Films to Chicken Meat

Silver migration was analyzed after chicken meat contact with the bio-AgNPs films. A commercial chicken was cut into pieces of approximately 1 g. Chicken meat pieces were involved by a film rectangle (2.5 cm × 4 cm) with and without bio-AgNPs and stored in the refrigerator (4–8 °C) for 1, 3, 5, 7, and 10 days or in the freezer (−20 °C) for 1, 3, 5, 7, 10 and 15 days. Each day had three samples of chicken in contact with films with and without bio-AgNPs. After contact with the films, the chicken meat pieces were dried at 60 °C for 48 h. The acidic digestion of dried chicken meat pieces was performed using an ultrapure Nitric Acid 5 N (Merck, Rahway, NJ, USA) at 60 °C for 48 h, according to Alves and Wood [71]. Silver in chicken meat was quantified through electrothermal atomic absorption spectrophotometry (AAnalyst 700, Perkin Elmer, Waltham, MA, USA).

### 5.8. Statistical Method

Analyses were performed using GraphPad Prism, version 9.1.1 (GraphPad Software), to the growth curve with bio-AgNPs films and silver migration to chicken meat. Means were analyzed using Student’s t-test. Values of *p* < 0.05 were considered significant.

## 6. Conclusions

This study’s biogenic and ecological silver nanoparticles showed great activity against a critical foodborne pathogen, *Salmonella* sp., including multidrug-resistant strains. This work showed the bacteriostatic effect of bio-AgNP films and how this affects bacterial kinetics. Although the results demonstrate high potential in using bio-AgNPs in biodegradable films to avoid food contamination, the silver migration in chicken cuts exceeded the limit proposed by the EFSA. Thus, considering that the release of silver from the film to chicken meat can be harmful due to its high concentration, it is necessary to improve the proposed technology to be used as a product.

## Figures and Tables

**Figure 1 antibiotics-12-00178-f001:**
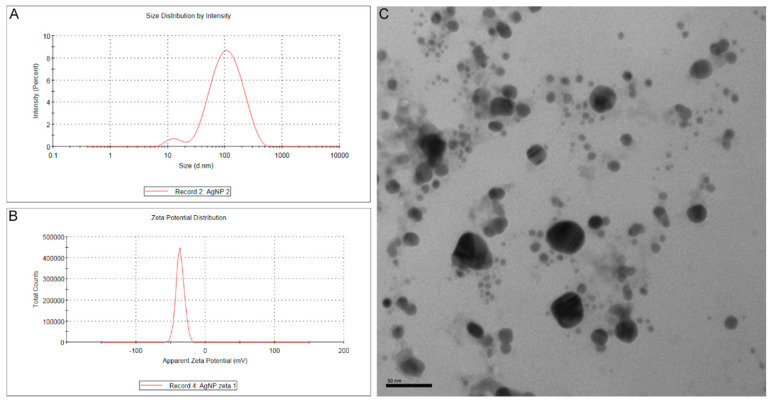
(**A**) Size distribution by intensity (%) of bio-AgNPs produced with *Fusarium oxysporum* components. The mean diameter of bio-AgNPs was 81.25 nm with a polydispersity index (PDI) value of 0.296. (**B**) Apparent zeta potential distribution by total counts of bio-AgNPs produced with *Fusarium oxysporum* components. The mean zeta potential of bio-AgNPs was −36.4 mV. (**C**) Transmission electron micrograph of bio-AgNPs. Spherical nanoparticles were observed in this micrograph (200,000×).

**Figure 2 antibiotics-12-00178-f002:**
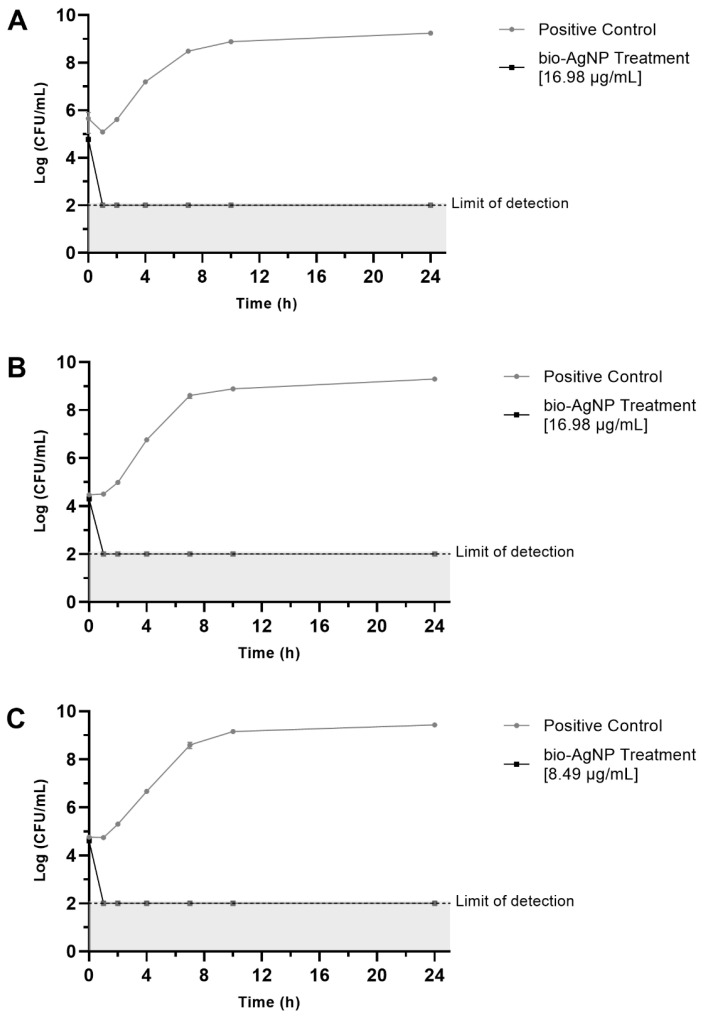
Time-kill curves of bacteria exposed to bio-AgNPs. Positive Control indicates bacterial growth without bio-AgNPs. The data are expressed as the mean ± standard deviation of three replicates. (**A**) *S.* Enteritidis ATCC 13076, (**B**) *S*. Typhimurium UK-1 ATCC 68169, and (**C**) *S.* Typhimurium ATCC 13311.

**Figure 3 antibiotics-12-00178-f003:**
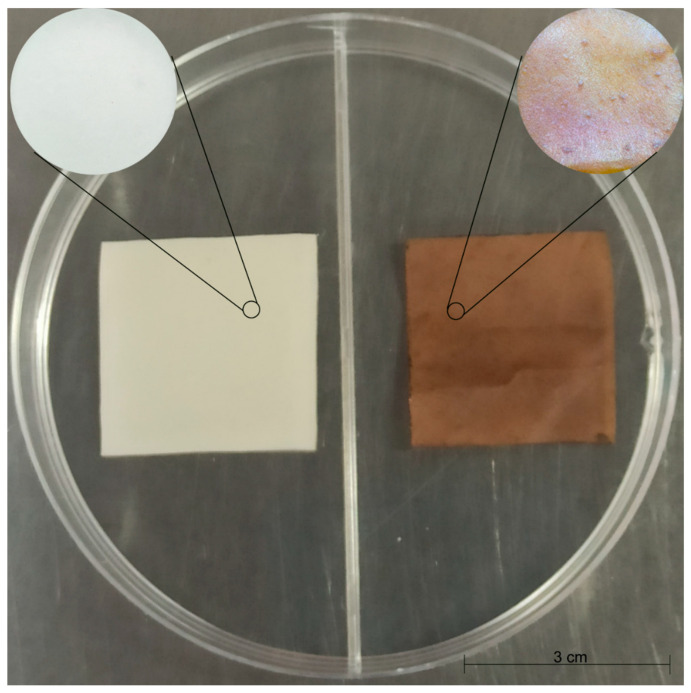
Biodegradable films with (brown) and without (white) bio-AgNPs. Film dimensions were 3 cm^2^. For the control film (white), 18.90% (*w*/*w*) of glycerol was used, while for the impregnated film (brown), 17.40% (*w*/*w*) of glycerol and 1.50% (*w*/*w*) of lyophilized bio-AgNPs were used.

**Figure 4 antibiotics-12-00178-f004:**
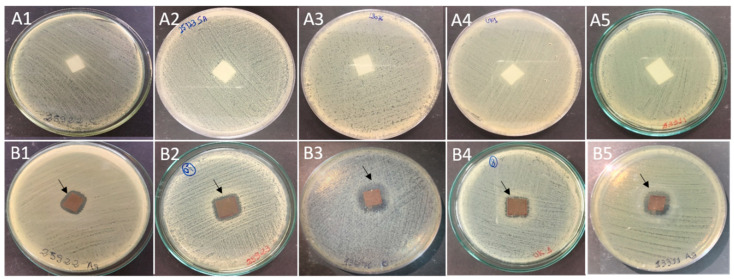
Antibacterial activity of films incorporated with bio-AgNPs indicated by agar diffusion assay. The bacteria growth was inhibited in the contact area with bio-AgNPs films, where an inhibition halo was observed (arrowheads). (**A**) Control (without bio-AgNPs) and (**B**) Treatment (with bio-AgNPs); 1: *E. coli* ATCC 25922; 2: *S. aureus* ATCC 25923; 3: *Salmonella* Enteritidis ATCC 13076; 4: *Salmonella* Typhimurium UK-1 ATCC 68169; 5: *Salmonella* Typhimurium ATCC 13311.

**Figure 5 antibiotics-12-00178-f005:**
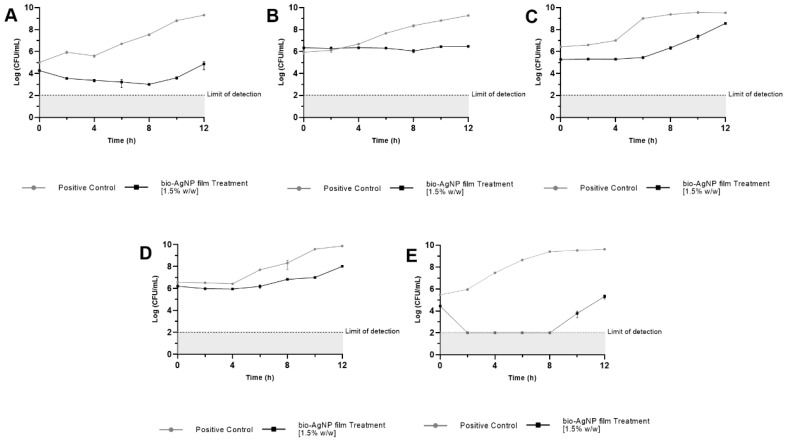
Growth curve of bacteria after 24 h of contact with biodegradable films with (treatment) and without bio-AgNPs (control). The data are expressed as the mean ± standard deviation of three replicates. (**A**) *E. coli* ATCC 25922, (**B**) *S. aureus* ATCC 25923, (**C**) *S.* Enteritidis ATCC 13076, (**D**) *S*. Typhimurium UK-1 ATCC 68169, and (**E**) *S*. Typhimurium ATCC 13311.

**Figure 6 antibiotics-12-00178-f006:**
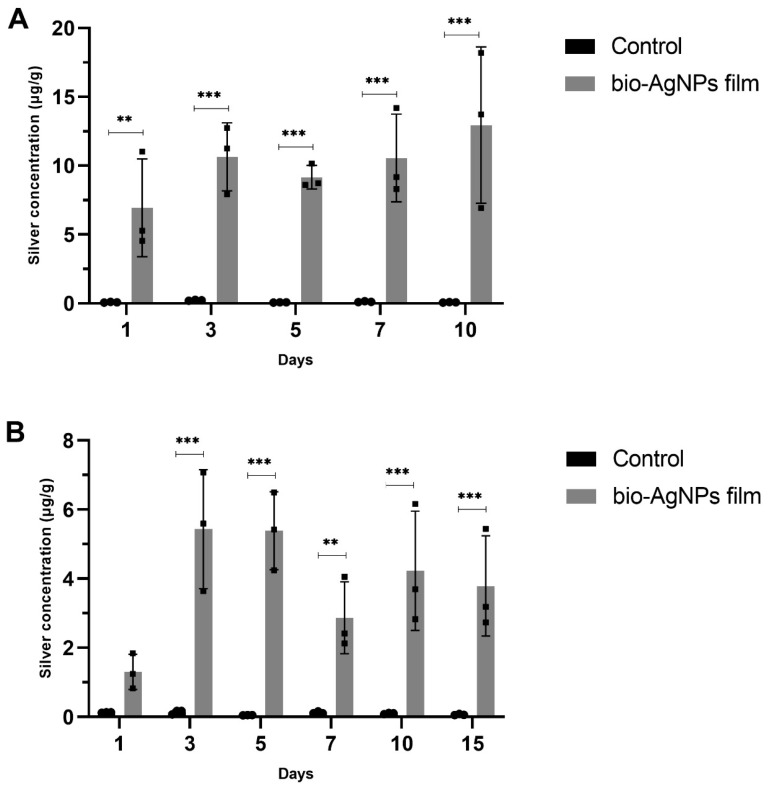
Quantification of silver migration from bio-AgNPs films to chicken meat pieces stored in the (**A**) refrigerator and (**B**) freezer. The data are expressed as the mean ± standard deviation of three replicates. Statistically significant differences between the groups were determined using an unpaired Student’s *t*-test. ** *p* < 0.01; *** *p* < 0.001 comparing control and bio-AgNPs film-treated groups. The control group indicates chicken pieces treated with films without bio-AgNPs.

**Table 1 antibiotics-12-00178-t001:** Values of MIC and MBC of bio-AgNPs.

Bacteria	MIC (µg/mL)	MBC (µg/mL)
*E. coli* ATCC 25922	4.24	4.24
*S. aureus* ATCC 25923	16.98	16.98
*S.* Enteritidis ATCC 13076	8.49	8.49
*S.* Typhimurium UK-1 ATCC 68169 *S.* Typhimurium ATCC 13311	8.49 4.24	8.49 8.49
*S*. Saint Paul 09	8.49	8.49
*S.* Saint Paul 10	8.49	8.49
*S.* Saint Paul 11	8.49	16.98
*S.* Saint Paul 12	16.98	16.98
*S.* Saint Paul 13	8.49	8.49
*S.* Enteritidis 92	8.49	8.49
*S.* Enteritidis 02	8.49	8.49
*S.* Enteritidis 06	8.49	8.49
*S.* Seftenberg 14	16.98	16.98
*S.* Seftenberg 15	16.98	16.98
*S*. Seftenberg 16	8.49	16.98
*S.* Seftenberg 17	8.49	8.49
*S*. Seftenberg 18	8.49	8.49
*S*. Seftenberg 19	8.49	16.98
*S*. Kentucky 20	16.98	16.98
*S*. Kentucky 21	8.49	8.49
*S*. Kentucky 22	16.98	16.98
*S*. Kentucky 25	16.98	16.98

MIC: Minimum Inhibitory Concentration; MBC: Minimum Bactericidal Concentration.

**Table 2 antibiotics-12-00178-t002:** Maximum concentration of bacteria (CFU/cm^2^) inhibited by bio-AgNP films.

Bacteria	Bacterial Growth		Maximum Bacterial Concentration Inhibited by Bio-AgNPs Films (CFU/cm^2^)
	Bio-AgNPs Films	Films without Bio-AgNPs	
*E. coli* ATCC 25922	−	+	10^5^
*S. aureus* ATCC 25923	−	+	10^5^
*S*. Enteritidis ATCC 13076	−	+	10^6^
*S*. Typhimurium UK-1 ATCC 68169	−	+	10^5^
*S*. Typhimurium ATCC 13311	−	+	10^6^

+ visible bacterial growth; − non-visible bacterial growth.

## Data Availability

The datasets used and analyzed during the current study are available from the corresponding author upon reasonable request.

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
