# Peer review of "Antibacterial Activity of Biodegradable Films Incorporated with Biologically-Synthesized Silver Nanoparticles and the Evaluation of Their Migration to Chicken Meat"

_antibiotics, 2023, doi:10.3390/antibiotics12010178_

Round 1
Reviewer 1 Report
In this paper, the authors developed starch‐poly films with silver nanoparticles produced with Fusarium oxysporum components (bio-AgNPs), the activity against different Salmonella serotypes to control foodborne pathogens and silver migration of these films were investigated. This research is interesting and meaningful, but several issues should be addressed before further evaluation for publication.
1. In figure 1, the size distribution and TEM seem uneven, the NPs couldn’t be concluded as spherical, and the SEM analysis of these NPs should be added.
2. What is the content ratio of AgNPs in the films?
3. In figures 2 and 5, the sample size (n) and error bar should be added to all of the curves.
4. As the author mentioned in lines 291-292, “the silver migration exceeds the limit”, is it possible to reduce the dose of bio-AgNPs to decrease the silver migration?
5. Line 335, “5.3. etermination” should be “determination”.
Author Response
Dear reviewer
All sugestions and corrections were replied in the attached file.
Thank you for the collaboration in this article.
Best regards.
Dr. Gerson Nakazato (Corresponding author)

Reviewer 2 Report
Overall, the study is well designed and evaluates the activity of Bio-AgNPs and a packaging film prepared thereof. There are study limitations, and the manuscript needs to be improved before being considered for publication.
1) there are many grammatical errors
2) there are many repetitions of results in the discussion section
3) conclusion is not well drawn and it needs to be amended
4) Since the packaging material is for food applications, cytotoxicity of the film or Bio-AgNPs need to be added to the manuscript
5) Discussion is not structured well
6) it is mentioned that experiments were performed in triplicate but results not showing that and there are no means and SD
Line 19 : replace minimal with a suitable alternative
Line 21: growth by upto" grammatical errors in sentence
Line 22: Add the allowed limit and give your results as well
Line 31: replace "Salmonella" with salmonellae
For more comments check the PDF file and improve the manuscript accordingly.

Author Response

(The authors gave the same response as above.)

Round 2
Reviewer 1 Report
The revised manuscript has improved a lot and could be accepted for publication.
Author Response
Dear reviewer
We thank you for all the comments and suggestions.
The corrections improved the quality of this manuscript.
Dr. Gerson Nakazato (Corresponding author)
Reviewer 2 Report
The revised manuscript reports the silver migration into chicken meat exceeding the limits set by EFSA. Conclusions are still not justified and toxicity studies are missing. Nanoparticles having antimicrobial activity have been reported previously in many studies. The films developed using these nanoparticles in this study are also antibacterial but their possible cytotoxicity and migration of silver into chicken meat are alarming which warrant further careful investigation before publication.
Author Response
Dear reviewer
We thank you for all the comments and suggestions.
We changed the title and added new comments in the abstract and conclusion section about high amount and migration of silver in chicken meat.
The corrections (rounds 1 and 2) improved the quality of this manuscript.
Response to Reviewer 2 Comments
Point 1: The revised manuscript reports the silver migration into chicken meat exceeding the limits set by EFSA. Conclusions are still not justified and toxicity studies are missing. Nanoparticles having antimicrobial activity have been reported previously in many studies. The films developed using these nanoparticles in this study are also antibacterial but their possible cytotoxicity and migration of silver into chicken meat are alarming which warrant further careful investigation before publication.
Response 1: According to the suggestions and the alteration of the discussion, we changed the title, so that it does not mislead the reader if he encounters the migration above the allowed limits. Furthermore, we made explicit the fact that the concentration of silver in chicken meat after the migration test is harmful and can be harmful. For this reason, we suggest the improvement of the technique in further studies.